# Optimisation of Biomass Production and Nutritional Value of Two Marine Diatoms (Bacillariophyceae), *Skeletonema costatum* and *Chaetoceros calcitrans*

**DOI:** 10.3390/biology11040594

**Published:** 2022-04-14

**Authors:** Carolina R. V. Bastos, Inês B. Maia, Hugo Pereira, João Navalho, João C. S. Varela

**Affiliations:** 1Centre of Marine Sciences, Campus de Gambelas, University of Algarve, 8005-139 Faro, Portugal; carolinavbastos@hotmail.com (C.R.V.B.); ibmaia@ualg.pt (I.B.M.); 2Necton S.A., Belamandil, 8700-152 Olhão, Portugal; info@necton.pt; 3GreenCoLab-Associação Oceano Verde, Campus de Gambelas, University of Algarve, 8005-139 Faro, Portugal; hugopereira@greencolab.com

**Keywords:** biomass growth, silica, nitrate, phosphorus, iron, micronutrients, *Skeletonema costatum*, *Chaetoceros calcitrans*

## Abstract

**Simple Summary:**

One of the key constraints that is associated with the production of microalgae biomass and products, is the low yields that are associated with high production costs in microalgae cultivation units. Therefore, the aim of the present work was to improve the biomass productivity of two high-value diatom species, *Skeletonema costatum* and *Chaetoceros calcitrans*. To do so, the culture medium that was supplied to the cultures was optimised in a stepwise process, regarding the nutrient’s silicate, nitrate, phosphorus, iron, and micronutrients. For both diatoms, the results that were obtained revealed a significant increase in biomass productivity as well as an improved biochemical profile regarding increased omega-3 fatty acids contents. With this work, the optimise culture media was established for each diatom, thus providing a strategy for lower production costs that were reflected in higher productivities with higher biomass quality. Ultimately this will help improve the application of *S. costatum* and *C. calcitrans* in the aquaculture and nutraceutical industries.

**Abstract:**

*S. costatum* and *C. calcitrans* are two cosmopolitan high-value centric diatoms, with a rich nutritional profile. The following work optimised the culture medium of *S. costatum* and *C. calcitrans* cultures, respectively, in a stepwise process as follows: 2.4 mM and 1.2 mM of silicate, 4 mM of nitrate, 100 µM of phosphate, 20 and 80 µM iron, and 0.5 mL L^−1^ of micronutrients. The results that were obtained revealed an increase in biomass productivity with a 1.8- and 3.2-fold increase in biomass that was produced by *S. costatum* and *C. calcitrans*, respectively. The biochemical profile showed an increase in high-value PUFAs such as 2.6-fold and 2.3-fold increase in EPA for *S. costatum* and *C. calcitrans*, respectively, whilst a 2.6-fold increase in DHA was detected in *S. costatum* cultures. The present work provides the basic tools for the industrial cultivation of *S. costatum* and *C. calcitrans* with enhanced productivity as well as improved biomass quality, two factors which are highly relevant for a more effective application of these diatoms to aquaculture and nutraceutical production.

## 1. Introduction

The optimisation of culture conditions is critical to improve the biomass productivities and nutritional value of microalgal cultures, since they are known to significantly affect growth and biochemical composition [1]. One method that is often employed to optimise growth in microalgae is the adjustment of nutrient supply [2]. For most microalgae, the proper supply of nitrates, phosphates, and iron are key factors that can readily be adjusted by the operator to promote optimal growth. However, in the case of diatoms, silica supplementation is essential to ensure optimal growth [3]. Besides having a crucial role in their natural habitats, the applicability of diatoms in human commercial activities can be vast. In recent years, their enormous economic potential has resulted in the generalised recognition of the biotechnological applications of diatoms [4]. In addition to their extensive use in aquaculture, the ability of diatoms to withstand highly polluted environments and the discovery of high-value metabolites in their biomass has led to novel emerging commercial applications in other areas such as: nitrogen-fixing biofertilizers, cosmetics, pharmaceuticals, renewable energy, health foods, fluid fuel production, raw materials production, wastewater detoxification, nanotechnology, and even in the field of forensic sciences [5,6,7,8,9,10]. Specifically, these applications are mainly related to their contents of silicate and lipids and their capacity to bioremediate pollutants through, for example, biosorption.

The optimisation of supplementation supply for high-value diatom species, such as *Skeletonema costatum* and *Chaetoceros calcitrans*, is crucial, given their outstanding potential for different biotechnological applications [11,12,13,14]. *S. costatum* is a diatom that often occurs in coastal waters and has the capacity to withstand fluctuating temperatures and salinity levels [15]. This strain forms straight chains that are linked by long, marginal, siliceous, tubular processes that can exceed the cell length [16]. Besides having a central nucleus, two chloroplasts can be observed, depending on the cell’s position (girdle view) [17]. In addition, it has also been reported to tolerate elevated nutrient levels such as nitrates, phosphates, and iron [18,19]. *S. costatum* extracts have also demonstrated antibacterial activity against several marine pathogenic bacteria [20,21]. Due to its rich nutritional profile, mainly in lipids (1.3–16.2% DW), proteins (23.3–31% DW), polyunsaturated fatty acids (PUFAs) such as EPA (6–23.5% of total FA) and DHA (1.41–4% of total FA) [15,22,23,24,25,26,27,28], *S. costatum* is presently commercialised as an aquaculture feedstock, more specifically as live feed in farming larviculture units for different molluscs and crustacean species [11]. In addition, extracts of this diatom species are also used in the cosmetic industry [12]. *C. calcitrans* is a diatom whose cells have been described as having four long setae on their corners whose poles join cells together and form chains [29]. This diatom has also been widely used as feed for larval, early juvenile, broodstock stages of bivalve molluscs, and as direct feed for crustaceans during early larval stages due to its balanced contents of PUFAs and vitamins [13]. This diatom has also proven its applicability in other fields, such as the development of novel drugs, given its reported fucoxanthin-rich fraction with effective anticancer properties [14]. 

Due to the rising interest in obtaining diatom-based bioproducts, more research regarding the optimal nutritional requirement for *S. costatum* and *C. calcitrans* growth and mass production has emerged [11,30,31,32]. Given this tendency alongside the ever-increasing potential of the marine biotechnology market, the future of the industrialisation of these cultures seems bright. The present study aimed at optimising biomass production and assessing the biochemical composition of two high-value diatoms: *S. costatum* and *C. calcitrans* through the fine-tuning of the nutrient supply, which is crucial for reduced production costs that are associated with higher productivities.

## 2. Materials and Methods

### 2.1. Culture Medium Optimisation 

*S. costatum* (Greville) Cleve and *C. calcitrans* (Paulsen) H. Takano inocula were obtained from Necton’s culture collection (SKC0119 and CHC0118, respectively). The culture medium optimisation trials were conducted in twelve 1-L bubble columns photobioreactors (PBRs) glass (Normax), whose diameter and light path were equal to the tubular PBRs that were used at Necton to simulate the industrial cultures’ growth (Figure 1A). Air flux was supplied at 800 mL min^−1^ filtered through 0.22 μm Whatman PTFE filters. CO_2_ was supplied when the cultures reached a pH higher than 8.5. The room temperature was set to 19 °C ± 1 °C. Each trial was carried out under a constant photosynthetic flux of 125 μmol photons m^−2^ s^−1^ with a duration of 7 days. The cultures were inoculated at the same initial optical density that was used in the company Necton S.A., which corresponded to distinct initial DWs of 0.18 ± 0.01 g L^−1^ for *S. costatum* and 0.12 ± 0.02 g L^−1^ for *C. calcitrans*, given the morphological differences between them. Through a fed-batch stepwise procedure, five parameters were tested in triplicate, regarding four concentrations, in the following order: (1) silicates: 0.4, 0.8, 1.2, and 2.4 mM; (2) nitrates: 1, 2, 4, and 8 mM; (3) phosphates: 50, 100, 200, and 400 µM; (4) iron: 10, 20, 40, and 80 µmol; and (5) micronutrients: 0.5, 1, 2, and 4 mL L^−1^. Nutribloom plus (NB^+^) was supplied as a control with final nitrate, phosphate, and iron concentrations of 4 mM, 200, and 40 µM, respectively. Micronutrients and vitamins were provided upon a 1:500 dilution of the respective stock solutions. Silicate was provided at a final concentration of 0.4 mM. The stepwise optimisation approach indicates that each nutrient was optimised one at a time, and the optimised value was used for the following trial. A schematic representation of the trials can be found in Figure 1B. The chosen order in nutrient optimisation is based on the existing know-how from previous results that were obtained by Necton S.A. The experience in growing marine diatoms from the lab to industrial scale show a common pattern, where the silicate concentration is always the most significant contributor followed by the macronutrients (nitrates and phosphorus) and iron, whereas the micronutrient solution commonly has a low effect (seawater commonly supplies all micronutrients used). Vitamins were not included in the present study, as their use in microalgal commercial scale cultivations is rare due to elevated costs. The composition of the culture medium NB^+^ is described in Table 1. 

### 2.2. Culture Monitoring

The culture growth was evaluated through optical density (OD) measurements using a UV-mini 240 UV/VIS spectrophotometer (Shimadzu, Kyoto, Japan) at a wavelength of 450 nm and DW determination through sample filtration, after which a calibration curve for growth was developed. For all the strains that were produced at Necton facilities, a scanning of wavelengths was conducted and compared to the traditional filtration method for dry weight determination, and the best wavelength was used for further works. For this reason, the wavelength of 450 nm was chosen given that this parameter gave a better correlation with the dry weight data that were obtained by filtration. The consumption of nitrate was determined by a modified Armstrong protocol [33]. Briefly, 0.5 mL of supernatant was added to a solution of 0.2 mL of HCl (83 mL L^−1^) and 9.3 mL of a NaCl solution (35.5 g L^−1^). The solutions were measured in a spectrophotometer at 220 nm and 275 nm. Silicate consumption was determined by a modified Smith and Milne protocol [34]. Briefly, a mixed reagent that was made of sulphuric acid and ammonium heptamolybdate tetrahydrate, and solutions of oxalic acid (100 g L^−1^) and ascorbic acid (17.5 g L^−1^) were added to 3 mL of supernatant. The samples were measured in a spectrophotometer at a wavelength of 810 nm. Phosphate consumption was determined by a test Kit Spectroquant^®^, where calibration curves were calculated with previously using spectrophotometric protocols.

### 2.3. Biochemical Analysis

The biochemical analysis was conducted in the final trial, and all cultures were harvested, centrifuged for 10 min at 2700× *g*, freeze-dried (LyoQuest Telstar, Terrassa, Spain), and stored at −20 °C. The protein content was evaluated by elemental analysis of C, H, and N through Vario EL III (Elementar Analysensysteme GmbH, Langenselbold, Germany), according to the manufacturer’s procedure. The total protein content was obtained by the multiplication of the percentage of N by the conversion factor of 4.78 and 4.63 for *C. calcitrans* and *S. costatum*, respectively [35]. The total lipids were determined according to Bligh and Dyer method [36], modified by Pereira [37]. The ash content was determined through gravimetry by burning 50 mg of the samples at 550 °C for 8 h, in a furnace. Carbohydrates were determined by the difference of the sum of total proteins, lipids, and ash content. The fatty acid profile was determined by using a modification of the Lepage and Roy protocol [38], described by Pereira [39].

### 2.4. Statistical Analysis

Statistical analyses were performed using IBM SPSS, version 26 (Armonk, New York, NY, USA: IBM Corp.) using Student’s *t*-tests and one-way ANOVA followed by Tukey tests (*p* < 0.05). All the trials were conducted with three biological replicates and all the analyses were carried out using three analytical replicates. The results are displayed as the mean ± standard deviation (*n* = 3).

## 3. Results and Discussion

### 3.1. Culture Optimisation

#### 3.1.1. Silicate Trials

In the first trial, higher growths were obtained with a silicate concentration of 2.4 mM, for both *S. costatum* and *C. calcitrans* (Figure 2A and Figure 3A).

For *S. costatum*, this growth was significantly higher than the remaining trials, with a final biomass concentration of 3.51 ± 0.17 g L^−1^ DW (*p* < 0.05; Figure 2A). Though, for *C. calcitrans*, this was not significantly higher than the growth that was obtained with a silicate concentration of 1.2 mM, where the final biomass concentration was of 2.03 ± 0.06 g L^−1^ DW (*p* > 0.05; Figure 3A). Hence, the silicate concentrations that were deemed optimal for *S. costatum* and *C. calcitrans* were 2.4 and 1.2 mM, respectively. From the literature, the maximum biomass concentration that was reported in *S. costatum* cultures was 1.52 g L^−1^ DW [40]. Whilst for *C. calcitrans*, Banerjee [41] reported a biomass concentration 2.20 g L^−1^ DW. For both diatoms, the lowest biomass was obtained under the control conditions (0.4 mM), with *S. costatum* and *C. calcitrans* reaching a final biomass concentration of 2.00 ± 0.03 and 0.76 ± 0.03 g L^−1^ DW, respectively. For this concentration, microscopic observations showed high cell death and cell elongation in both diatoms, the latter being more prominent for *S. costatum* (Figure 4). This could be explained by the fact that silicate limitation induces cell cycle arrest at the G1/S boundary and the transition of G2/M in diatom cultures as reported by Vaulot [42] and Brzezinski [43]. Furthermore, significant depletion of silicate was discovered in all the treatments, which could be explained by the fact that diatoms contain specialized silicate deposition vesicles (SDV), where the absorbed silicate is transported and stored for later use [44]. However, no studies were found regarding a similar silicate supply as well as an increase in biomass production such as the ones that are described in this study. In fact, for *S. costatum* cultures, 0.11 mM is the standard concentration that is applied to the culture media [15,45,46]. For *C. calcitrans* cultures, a silica supply of 0.2–0.4 mM is usually used [47,48]. Thus, these results underline the relevance of a significant increase in the silicate concentration for cultures with a continuous light source.

#### 3.1.2. Nitrate Trials

In the nitrate concentration optimisation trial, higher growth was obtained with a nitrate supply of 8 mM for *S. costatum*. Still, this growth was not significantly higher than that which was obtained with 4 mM, where the biomass concentration was of 3.18 ± 0.07 g L^−1^ DW (*p* > 0.05; Figure 2B). For *C. calcitrans*, a significantly higher growth was obtained for 4 mM of nitrate (*p* < 0.05), with a maximum DW of 2.17 ± 0.02 g L^−1^ (Figure 3B). For lower concentrations of nitrates, increased cell death was registered from the third trial day onwards, in both diatom cultures. For *C. calcitrans*, a significant increase in ‘mucilage’ attached to the cells and in the medium was detected. This could result in an increase in the total carbohydrates accompanied by the release of free extracellular polysaccharides and sugar-containing compounds that became bound to the cell surface, due to nitrate-deficient conditions [47]. These results were concurrent with the premise that a low supply of nitrate does significantly affect and stress microalgal cultures, thus being commonly used to increase the production of specific bioproducts, such as lipids [15]. Ultimately, for *S. costatum* and *C. calcitrans*, 4 mM was deemed to be the optimal nitrate supply. From what could be found in the literature, the concentrations of nitrate that was supplied in cultures of *S. costatum* and *C. calcitrans* are far below the concentration that is considered as optimal for these diatoms, with values ranging between 0.2–0.8 mM and 0.4–2 mM, respectively [15,47,48].

#### 3.1.3. Phosphate Trials

Regarding phosphate optimisation, for both diatoms, no significant growth differences were detected between the treatments (*p* > 0.05; Figure 2C and Figure 3C). Nonetheless, considering cell fitness and culture health that were evaluated through microscopic observations of the cellular morphology and the formation of chains, the supply of 100 µM phosphate was recognized as the optimal concentration for *S. costatum* and *C. calcitrans*, where the biomass concentrations of 3.72 ± 0.07 and 2.44 ± 0.02 g L^−1^ DW were obtained, respectively. For all the treatments, a high phosphate depletion was detected. The accumulation of polyphosphate as dense calcium-associated inclusions (e.g., acidocalcisomes) has previously been observed in *S. costatum* [49,50]. Hence, the mechanism that is known as “phosphorus luxury uptake” might justify the drastic depletion of phosphate from the culture medium, in addition to the possible precipitation of phosphorus with cations such as Ca^2+^ [51,52]. Regarding the data that are available in the literature, in *S. costatum* studies there is a dearth of reports regarding the optimal phosphate concentration, with a higher focus on the limiting effects of such a supply, where for example, Monkonsit [23] provided phosphate-limiting conditions to the cultures, with a supply of 16.8 µM. For *C.calcitrans* the standard phosphate concentration reported is usually circa 200 µM [48,53].

#### 3.1.4. Iron Trials

From the iron optimisation trial, for *S. costatum*, no significant differences were detected between the treatments (*p* > 0.05; Figure 2D). Yet, the concentration of 20 µM was selected as optimal from microscopic analysis, with a final biomass concentration of 2.60 ± 0.10 g L^−1^ DW. Conversely, for *C. calcitrans*, significantly faster growth was obtained at 80 µM, leading to a maximum biomass of 2.10 ± 0.02 g L^−1^ DW (Figure 3D). No similar supplies were detected in the literature for *C. calcitrans* cultures. Depending on the species, iron uptake and further luxury uptake in diatoms have been associated with the protein ferritin and/or storage vacuoles [54]. This superior storage ability of iron and other nutrients coincides with the bloom-forming capacity that diatoms are known to possess in natural-forming phytoplankton communities. Hence, these results shed some light on the potential use of *C. calcitrans* in metal bioremediation.

#### 3.1.5. Micronutrients Trials

In the optimal dilution of micronutrients trial for *S. costatum* and *C. calcitrans,* no significant differences were detected between the treatments (*p* > 0.05; Figure 2E and Figure 3E), where a supply of 0.5 mL L^−1^ was considered as optimal, with a maximum biomass concentration of 3.09 ± 0.07 and 1.99 ± 0.05 g L^−1^ DW, respectively. From this trial, it was possible to conclude that the micronutrient solution, mainly composed of magnesium, zinc, molybdenum, and manganese, is only required in trace amounts. From previous studies, it was possible to see that there is a lack of research regarding the effects of micronutrients on the growth of *S. costatum* and *C. calcitrans*. However, the evaluation of the effects of zinc and cobalt on microalgal growth is recurrent in different studies where usually their toxicity is tested [55,56]. The growth performances of both diatoms on table format is also available in the Appendix A. The tested nutrient combinations throughout the stepwise optimisation approach, and the final optimised results that were obtained for each diatom species are summarized in Table 2 (*S. costatum*), and Table 3 (*C. calcitrans*).

### 3.2. Biochemical Analysis

#### 3.2.1. Proximal Composition

Regarding *S. costatum*¸ throughout the nutrient optimisation process the most noticeable modification of the proximal composition of the cultures was observed in the ash content (Figure 5A), which in comparison to the control conditions, 28.7 ± 0.6% of DW, significantly increased from the silicate trial onwards, where a maximum of 52.5 ± 1.5% of DW was detected in the iron trial (*p* < 0.05). The same trend, on the other hand, was not observed for *C. calcitrans*, where an increase in the ash content was not detected throughout the optimisation trials (Figure 5B), with values ranging between 45.4 and 49.4 % of DW, in comparison to the control conditions (50.5 ± 0.6% of DW). The results that were obtained with *S. costatum* may be explained by the increased silicification of the cell walls, due to the increase in the silicate concentration present in the medium [57]. Furthermore, although there is a general shortage of information available regarding the ash content in both *S. costatum* and *C. calcitrans* cultures, a study by Lestari [24] reported an ash content of 55.6% DW for *S. costatum*, whilst for *C. calcitrans* Kudaibergenov and Khajiyeva [58] reported an ash content of 19.5% DW. The present work demonstrates the high nutrient storage capacity of diatoms, as well as an elevated rate of cell wall silicification, which could explain the elevated ash contents previously described in *S. costatum* cultures [59,60,61]. 

Regarding other cellular components, although no significant modifications in the proximal composition were detected for *C. calcitrans*, the opposite trend was detected in *S. costatum* cultures, regarding protein and carbohydrates content (Figure 5A). The silicate and nitrate optimisation trials resulted in no significant changes in the protein content in comparison to the control conditions (19.6 ± 0.5%, 19.7 ± 0.1%, and 18.8 ± 1.2% of DW, *p* > 0.05). However, a significant decrease in the protein content was detected from the phosphate trial onwards (*p* < 0.05), where the protein content ranged between 14.3–15.8% of DW. These results suggest that a phosphate concentration shift from 200 to 100 μM (optimal conditions for growth) negatively affected the protein content in *S. costatum* cultures. Taking into consideration that phosphorus is an integrating nutrient in the biosynthesis of photosynthesis-related proteins and ATP (adenosine triphosphate), the decrease in protein content might be explained by a reduction in ATP synthesis, resulting consequently in the deficiency of available energy for the assimilation of nitrogen, which is essential for protein synthesis [62,63]. In addition, the maximum values of the total protein content that were obtained in *S. costatum* cultures, although slightly lower, were similar to the ones that were found in the literature, where the protein content ranged between 23.3–31% DW [22,23,24,25]. For *C. calcitrans*, the protein contents ranged between 15.3–16.9% DW. These results differed from what has been reported in the literature, where the total protein content ranged between 31.3–41.6% DW [22,23,41,64].

For *S. costatum* under control conditions, a higher percentage of carbohydrates were detected (34.1 ± 0.3% of DW, *p* < 0.05), which decreased in the nitrate, phosphate, iron, and micronutrients trials where the values ranged between 15.8–26.2% of DW (Figure 5A). These results may indicate that the cellular carbohydrates were converted into other biochemical constituents [65]. In addition, we suggest that the higher content of carbohydrates that were observed in the control conditions could be associated with the lower growth that was registered and the harvesting of cultures in the stationary phase. Under these conditions, an increase in the storage and cell wall polysaccharides occurs due to a lower rate of cell division, since structural polysaccharides are crucial constituents in the organic casing of the siliceous components of the diatom cell [66,67]. The carbohydrate content that was detected in this work for *S. costatum* was similar to what is found in the literature, where values ranged between 4.6–26.4% DW [22,23,25]. The same trend was observed for *C. calcitrans*, whose carbohydrate contents that were detected (23.4–26.1% DW), were consistent with what has been described in the literature where the values range between 6–37% DW [22,23,41,64]. Overall, the integration of growth data with proximal composition demonstrates two outcomes: altering the culture medium composition for *S. costatum* highly influenced both culture growth and the proximal composition of the cultures, and for *C. calcitrans*, this optimisation of the culture medium had only a more pronounced effect on culture growth itself (Figure 5).

#### 3.2.2. Fatty Acid Profile of Control and Optimised Conditions

For the control conditions of *S. costatum* and *C. calcitrans*, lower contents of eicosapentaenoic (EPA) and docosahexaenoic (DHA) acids were detected. *S. costatum* had an EPA and DHA content of 8.9 ± 5.6% and 2.0 ± 1.2% of total fatty acids (TFA), respectively, whereas *C. calcitrans* had a content of 10.5 ± 0.6% and 1.3 ± 1.2% of TFA in EPA and DHA, respectively (Table 4). For the optimised conditions, however, higher EPA contents were obtained (*p* < 0.05) in *S. costatum* (22.2 ± 1.3% of TFA) as well as *C. calcitrans* (18.8 ± 2.3% of TFA, Table 4). For *S. costatum*, the DHA content was significantly higher than that of the control conditions (5.2 ± 0.3% of TFA, *p* < 0.05, Table 4). Though, for *C. calcitrans*, no significant differences were detected between the DHA contents of optimised and control conditions (1.3 ± 0.2% and 1.9 ± 0.2% of TFA, respectively; Table 4). In general, EPA (C20:5) is normally found in *S. costatum* with significant values ranging between 6–23.5% of TFA, while DHA (C22:6) is reportedly found at lower values, ranging between 1.41–4% of TFA [15,25,26,27,28]. For *C. calcitrans*, the higher EPA contents compared to the DHA levels is in accordance with values that were obtained previously, which range between 5–26.3% of TFA and 0.7–2.3% of TFA, respectively [26,31,53,68,69].

Regarding other FA, the profiles of both diatoms were mainly composed of palmitoleic (C16:1), myristic (C14:0), palmitic (C16:0), hexadecatrienoic (C16:3), and stearidonic (C18:4) acids, with *S. costatum* also containing stearidonic (C18:4) acid (Table 4). Additionally, similarly to what was obtained in the present work for *S. costatum* cultures, other FA have been reported with interesting values, such as myristic (14.1–31.79% of TFA), palmitic (10.7–45.36% of TFA), palmitoleic (14.78–31.15% of TFA), hexadecatrienoic (3.7–16.3% of TFA), and stearidonic (3.4–4.4% of TFA) acids [15,25,26,27,70,71]. Moreover, although Van Houcke [25] reported 11.2 ± 0.6% of TFA of palmitidonic acid (C16:4) for *S. costatum* cultures, we detected trace amounts not only in the control conditions (0.6 ± 0.3% of TFA) but also for the remaining trials where the values ranged between 0.6–2.2% of TFA (Table 4). For *C. calcitrans,* other FA have also been reported in interesting amounts, such as myristic (9.2–23.2% of TFA), palmitic (10.7–29.7% of TFA), palmitoleic (17.4–32.5% of TFA), and hexadecatrienoic (6.4–14.5% of TFA) acids, although there are reports of stearic (5.4–16.7% of TFA) and elaidic (9.8–12.1% of TFA) FA [26,31,53,68,69,72]. In this work, however, these FA were only detected in trace amounts with values ranging between 0.9–1.6% of TFA and 1.7–4.6% of TFA, respectively (Table 4).

The stepwise optimisation process resulted in the increase of EPA and DHA contents, the latter in the case of *S. costatum*, where a marked increase was obtained with the optimisation of the concentrations of silicates and nitrates until the final trial, where the composition of the culture medium NB^+^ was optimal for the growth of both diatoms. The production of these FA is highly relevant given their essential role in human and animal nutrition [39,73]. Therefore, these results establish the high-value potential of these strains, given the increasing need for n-3 PUFA-rich feedstocks for the aquaculture and nutraceutical industries [74,75].

Under control conditions, the PUFA content of *S. costatum* and *C. calcitrans* was significantly lower (*p* < 0.05), 21.6 ± 11.1% of TFA and 17.7 ± 0.3% of TFA, respectively, when compared with the optimised conditions, 56.2 ± 1.4% of TFA and 29.9 ± 0.5% of TFA, respectively (Table 4). In stressful environmental conditions (i.e., nutrient limitation), cells favor the biosynthesis of SFA or MUFA in detriment to the biosynthesis of PUFA, since SFA and MUFA can store more energy and allow the cells to maintain homeostasis [15,76]. In addition, the same FA adjustment has been described for cultures that were collected in the stationary growth phase [77]. These results suggest an adaptive strategy where, in the control cultures, the FA profile was shifted to a higher content of storage lipids (SFA and MUFA) as a response mechanism to the silica-deficit conditions, clearly indicating that the cultures of both diatoms were under stress. From the available literature, for *S. costatum*, the total SFA ranged between 17.3–73.6% of TFA, the total MUFA ranged between 14.8–36.6% of TFA, and the total PUFA ranged between 9.9–61.7% of TFA [15,25,26,27,28,70]. Whilst for *C. calcitrans*, according to the literature, the total saturated (∑SFA), monounsaturated (∑MUFA), and polyunsaturated (∑PUFA) FA ranged between 11.7–50.2% of TFA, 19.6–39.1% of TFA, and 5.0–50.9% of TFA, respectively [26,31,53,68,69,72].

## 4. Conclusions

The present work significantly improved the basal medium that is used by Necton, from which vital nutrients were almost fully optimised. Under laboratory conditions, the key step to enhance the biomass production of continuously illuminated *S. costatum* and *C. calcitrans* cultures was, respectively, a six- and three-fold increase in the silicate supply. Besides silicate, only nitrate-deficient conditions resulted in growth arrest, with the already established supply proving to be optimal. Regarding the optimisation of the supply of phosphate, iron, and micronutrients, it is possible to assume that they have a smaller impact on growth. We can conclude that the supply of silicate and nitrate are key factors, at the nutrient level, that can be optimised to increase biomass production for both diatoms. Overall, the culture medium optimisation led to a maximum 1.8-fold and 3.2-fold increase in the biomass that was produced by *S. costatum* and *C. calcitrans* cultures, respectively. Furthermore, a shift in the fatty acid composition was observed under which the control cultures contained higher amounts of MUFA and SFA than PUFA, since they were under nutrient-stress conditions. The increment in EPA and DHA was detected and associated with improved culture growth, an essential observation to maximize both biomass and PUFA productivities. Taken together, these results strongly suggest that the stepwise nutrient optimisation process that is described here is of the utmost importance for the industrial production of these high-value diatom species, following the paradigm of increased productivity with reduced associated costs, two essential factors for the success of any industry. Ultimately, this work is crucial to assist numerous production units, which cannot currently cultivate these species due to unsustainable investment requirements.

## Figures and Tables

**Figure 1 biology-11-00594-f001:**
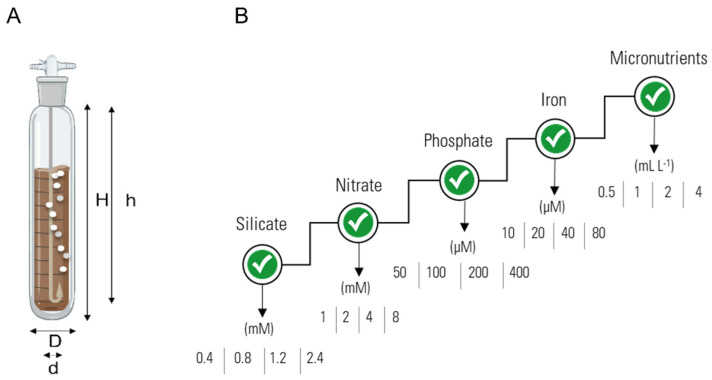
(**A**) schematic representation and dimensions of the bubble columns PBRs system that was used in nutrient optimisation trials. H, column height: 43 cm; h, draft tube height: 41 cm; D, column diameter: 6 cm; d, draft tube diameter: 1 cm. (**B**) schematic representation of the stepwise optimisation process that was applied in the present work. Each nutrient was optimised individually regarding four different concentrations. The optimised concentration was employed in the following step of nutrient optimisation.

**Figure 2 biology-11-00594-f002:**
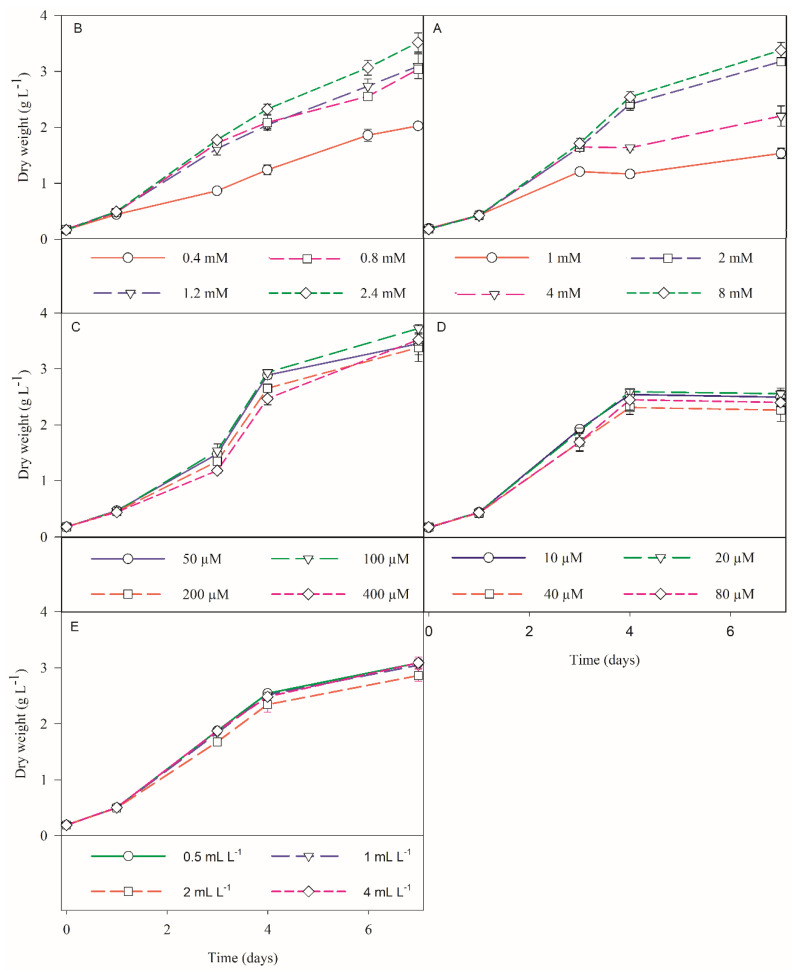
*S. costatum* growth performance in terms of biomass dry weight (g L^−1^) that was obtained in 1-L bubble column PBRs in order to optimise the supply of a specific nutrient (**A**) silicate, (**B**) nitrate, (**C**) phosphate, and (**D**) iron or (**E**) micronutrient concentrations (*n* = 3). The values are represented as the mean ± standard deviation.

**Figure 3 biology-11-00594-f003:**
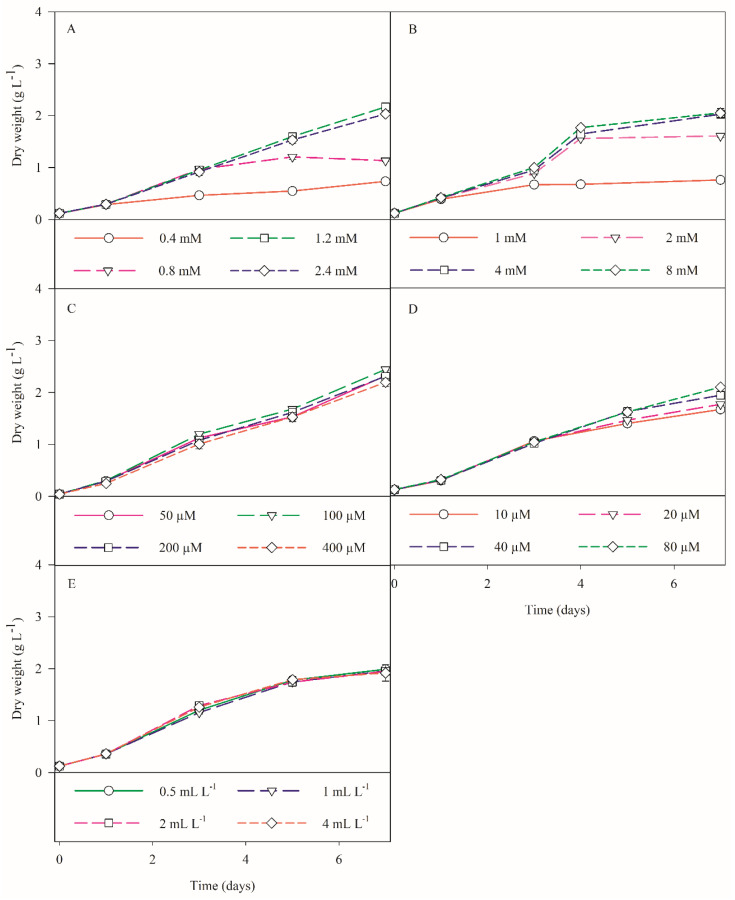
*C. calcitrans* growth performance in terms of biomass dry weight (g L^−1^) that was obtained in 1-L bubble column PBRs in order to optimise the supply of a specific nutrient (**A**) silicate, (**B**) nitrate, (**C)** phosphate, and (**D**) iron or (**E**) micronutrient concentrations (*n* = 3). The values are represented as the mean ± standard deviation.

**Figure 4 biology-11-00594-f004:**
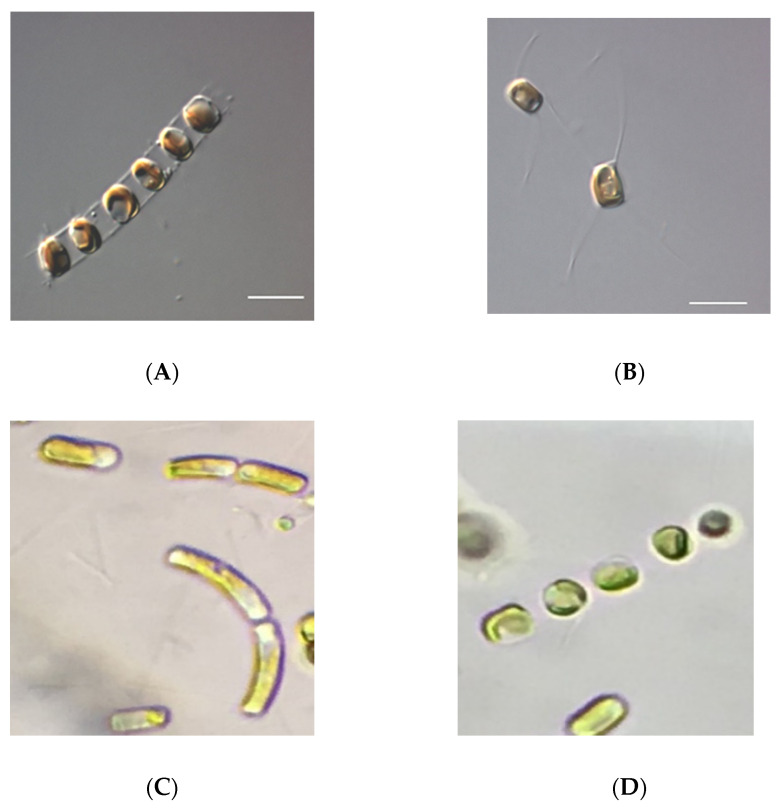
Microscopic observations of diatoms: *Skeletonema costatum* (**A**), *Chaetoceros calcitrans* (**B**), and *Skeletonema costatum* cells growing with a silicate supply of 0.4 mM (**C**), where unhealthy elongated cells were observed, and *Skeletonema costatum* cells growing with a silicate supply of 2.4 mM (**D**), where the cultures had a healthy morphology. For (**A**,**B**), the differential interference contrast (DIC) and a 100× lens with additional 1.6× amplification via an Optovar module was used. Scale bar = 20 µm.

**Figure 5 biology-11-00594-f005:**
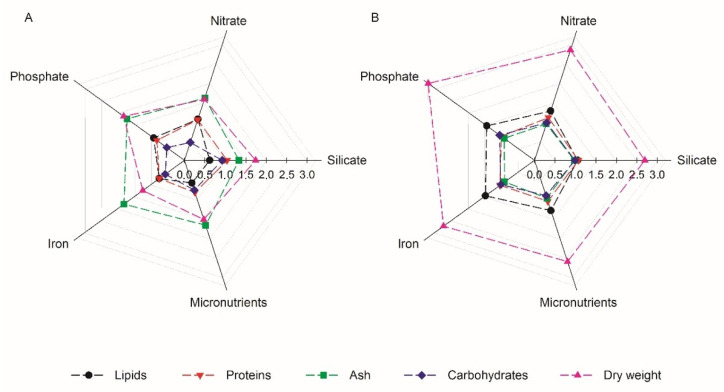
Proximal composition of optimised biomasses of diatoms *S. costatum* (**A**) and *C. calcitrans* (**B**) that were grown in bubble column PBR system, regarding the dry weight as well as the total protein, lipid, ash, and carbohydrate contents (*n* = 3). The values are represented as the means ratios to the control conditions.

**Table 1 biology-11-00594-t001:** Elemental composition of the culture medium Nutribloom plus.

Macronutrients	Metals	Vitamins
**2 M N**	20 Fe (mM)	10.4 Thiamine (µM)
**100 mM P**	1 Zn (mM)	2 Biotin (µM)
	1 Mo (mM)	2 B12 (µM)
	1 Mn (mM)	
	2 Mg (mM)	
	0.1 Cu (mM)	
	0.1 Co (mM)	

**Table 2 biology-11-00594-t002:** Nutrient combinations as well the optimised concentrations that were obtained throughout the stepwise nutrient optimisation process of the *S. costatum* growth medium.

Trial	Treatments	Silicate (mM)	Nitrate (mM)	Phosphate (µM)	Iron (µM)	Micronutrients (mL L^−1^)
**Silicate**	Tested nutrients	0.4|0.8|1.2|2.4	4	200	40	2
	Optimised combination	2.4	4	200	40	2
**Nitrate**	Tested nutrients	2.4	1|2|4|8	200	40	2
	Optimised combination	2.4	4	200	40	2
**Phosphate**	Tested nutrients	2.4	4	50|100|200|400	40	2
	Optimised combination	2.4	4	100	40	2
**Iron**	Tested nutrients	2.4	4	100	10|20|40|80	2
	Optimised combination	2.4	4	100	20	2
**Micronutrients**	Tested nutrients	2.4	4	100	20	0.5|1|2|4
	Final optimised combination	2.4	4	100	20	0.5

**Table 3 biology-11-00594-t003:** Nutrient combinations as well the optimised concentrations that were obtained throughout the stepwise nutrient optimisation process of the *C. calcitrans* growth medium.

Trial	Treatments	Silicate (mM)	Nitrate (mM)	Phosphate (µM)	Iron (µM)	Micronutrients (mL L^−1^)
**Silicate**	Tested nutrients	0.4|0.8|1.2|2.4	4	200	40	2
	Optimised combination	1.2	4	200	40	2
**Nitrate**	Tested nutrients	1.2	1|2|4|8	200	40	2
	Optimised combination	1.2	4	200	40	2
**Phosphate**	Tested nutrients	1.2	4	50|100|200|400	40	2
	Optimised combination	1.2	4	100	40	2
**Iron**	Tested nutrients	1.2	4	100	10|20|40|80	2
	Optimised combination	1.2	4	100	80	2
**Micronutrients**	Tested nutrients	1.2	4	100	80	0.5|1|2|4
	Final optimised combination	1.2	4	100	80	0.5

**Table 4 biology-11-00594-t004:** Fatty acid profiles of *S. costatum* and *C. calcitrans* from the control (C) and optimised (O) trials (*n* = 3). The values are expressed as the mean of total FAME percentages ± standard deviation.

Fatty Acids (%)	*Skeletonema costatum* (C)	*Skeletonema costatum* (O)	*Chaetoceros calcitrans* (C)	*Chaetoceros calcitrans* (O)
(C14:0)	31.4 ± 3.8	17.3 ± 0.3	15.8 ± 1.2	13.1 ± 1.1
(C16:0)	7.3 ± 1.4	4.9 ± 0.0	32.9 ± 1.1	20.6 ± 1.6
(C18:0)	n.d.	n.d.	1.6 ± 0.4	1.4 ± 0.1
∑SFA	36.3 ± 7.4 ^a^	18.9 ± 2.8 ^b^	50.3 ± 0.5 ^a^	34.4 ± 0.8 ^b^
(C16:1)	40.9 ± 2.9	23.6 ± 1.4	27.2 ± 1.0	32.9 ± 0.7
(C18:1)	2.6 ± 0.5	1.3 ± 0.0	4.6 ± 0.6	2.8 ± 0.5
∑MUFA	42.2 ± 4.3 ^a^	24.9 ± 1.4 ^b^	31.8 ± 0.6 ^b^	35.7 ± 0.6 ^a^
(C18:4)	0.6 ± 0.3	2.2 ± 0.1	n.d.	n.d.
(C16:4)	6.6 ± 2.4	20.1 ± 0.5	n.d.	n.d.
(C16:3)	3.2 ± 1.3	6.5 ± 0.1	6.1 ± 0.5	9.2± 2.6
(C20:5)	8.9 ± 5.6	22.2 ± 1.3	10.5 ± 0.6	18.8 ± 2.3
(C22:6)	2.0 ± 1.2	5.2 ± 0.3	1.3 ± 0.2	1.9 ± 0.2
∑PUFA	21.6 ± 11.1 ^b^	56.2 ± 1.4 ^a^	17.7 ± 0.3 ^b^	29.9 ± 0.5 ^a^

SFA: Saturated fatty acids; MUFA: Monounsaturated fatty acids; PUFA: Polyunsaturated fatty acids; n.d.: not detected; ^a,b^: Different letters denote significant differences between conditions (*p* < 0.05).

## Data Availability

Data available on request.

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
