# Peer review of "Optimisation of Biomass Production and Nutritional Value of Two Marine Diatoms (Bacillariophyceae), Skeletonema costatum and Chaetoceros calcitrans"

_biology, 2022, doi:10.3390/biology11040594_

Round 1

Reviewer 1 Report

The authors provide a report on growing two species of diatoms in the lab, and what might help their growth to optimal levels. Unfortunately, the study is not justified- as a plethora of literature on the subject (dating back decades) exists, and the authors do not present a rational for conducting this work, or why these particular two species (out of many) were selected. The manuscript is devoid of all photos, which is unfortunate as they do describe some apparent interesting morphological variations in their lab diatoms under different stresses- but unfortunately these are not shown. Not only should these be shown rather than discussed, but basic micrographs for these species should be presented.

The paper also has no morphological information about these species (indeed even their habitat is lacking) and how they fit into a given food chain. Only a handful of sources are used to show how diatoms might be important in other areas, despite them being extremely important to ecosystem health and immerging biotechnology.

I am unconvinced this data set is novel or robust enough to be considered for publication. Although much data is presented, the rational behind it is completely lacking. The experiment also has several flaws- the first being the constraints of an optimal lab setting which may have 0 effect in an aquaculture setting. Further methodological issues should be examined by the other reviewers.

Some specific examples are:

L69: The study used triplicate examinations of a 1L test system. These numbers are far too low to extrapolate anything on a commercial aquaculture scale (e.g. factors of magnitude higher).

L81: optics were used for culture growth calculations- no live observations were done? There are numerous issues with this omission. L116-117 says microscopy was done, which showed cell death and elongation- so the density of healthy diatoms could be far lower than presented here.

The conclusions are less than 20 lines, which is far too short. Perhaps this is indicative of the study not being able to conclude anything novel from the simple design?

The paper is in perfect British English. The figures are well put together, but the text is confusing jumping between the two species. The choices of what nutrients to include and to what level to vary them are also missing within the text, despite a list of 45 references.

Due to the issues with the methods, rational of the overall study, and lack of novelty (such that most of this work has certainty already been done after a simple google search) I do not feel that this work is suitable for publication in this journal, despite the topic potentially being very helpful to other researchers in the field and overall aquaculture.

Author Response

Reviewer #1:

The authors provide a report on growing two species of diatoms in the lab, and what might help their growth to optimal levels. Unfortunately, the study is not justified- as a plethora of literature on the subject (dating back decades) exists, and the authors do not present a rational for conducting this work, or why these particular two species (out of many) were selected. The manuscript is devoid of all photos, which is unfortunate as they do describe some apparent interesting morphological variations in their lab diatoms under different stresses- but unfortunately these are not shown. Not only should these be shown rather than discussed, but basic micrographs for these species should be presented.

Answer:  The authors disagree with the reviewer’s comment since the data on the nutritional needs of both diatoms is definitely scarce in the literature. Although there is wide literature in ecology works regarding diatoms and the nutrients present in their ecosystems, specific works targeting the optimal nutrient requirements are scarce or are quite outdated. This fact is highlighted by the high biomass concentrations obtained in this work, which have never previously been reported. Furthermore, it is widely established that microalgae display quite different nutrient needs among species of the same genus and even among strains, and the present works focuses on establishing the optimum nutrient needs, a step that is essential for the industry to improve large-scale production of biomass. The rationale of the work is simple and is related to understanding the nutritional needs of both diatoms and establishing an optimised cultures medium for industrial ventures. Therefore, the introduction was amended to better highlight the rationale behind the work conducted.

The justification of choosing these two species is present in the introduction and is fully related to the high-value and market demand for biomass of both diatoms, which is currently applied to commercial sectors such as aquaculture and cosmetics. However, their cultivation at large scale is challenging (based on the results obtained by Necton S.A. in the last decade), and significant improvements are needed, including the improvements made in the culture medium used for biomass production, a direct result of this work.

Lastly, we agree with the reviewer and photos of both strains were added to the manuscript to better illustrate the manuscript.

The paper also has no morphological information about these species (indeed even their habitat is lacking) and how they fit into a given food chain. Only a handful of sources are used to show how diatoms might be important in other areas, despite them being extremely important to ecosystem health and immerging biotechnology.

 Answer: The lack of morphological information regarding these species is justified by the scarce presence of such studies in the literature. Information regarding their habitats and application in different areas is present in the introduction, which was further extended as suggested by the editor and reviewers to improve this information in the manuscript.

I am unconvinced this data set is novel or robust enough to be considered for publication. Although much data is presented, the rational behind it is completely lacking. The experiment also has several flaws- the first being the constraints of an optimal lab setting which may have 0 effect in an aquaculture setting. Further methodological issues should be examined by the other reviewers.

 Answer: Again, the authors disagree with the reviewer. These data is novel regarding the biomass concentrations that we were able to achieve with these species; no similar values have been published in the literature nor were previously achieved by the company Necton S.A. Furthermore, this work is the baseline tool to help companies such as Necton, decrease the production costs associated with the cultivation of these species, through an optimised culture medium. It will help other companies overcome their constraints to also grow these strains successfully. As previously mentioned, the manuscript text was improved to better describe the rationale underlying the work carried out and reported in this manuscript.

In more than two decades growing microalgae from lab to industrial scale, our experience is that the optimisation of culture medium at laboratory scale is essential and brings significant advantages to large scale production. In fact, the bubble columns used in this work have exactly the same diameter and light path as the tubular photobioreactors used at Necton to simulate the industrial-scale culture growth. We cannot understand how the optimisation of a culture medium could be performed in any other way, since a similar work would be completely impossible to be conducted in 20 m3 photobioreactors and environmental conditions changing daily. Our strong conviction is that to efficiently perform an optimisation study and understand the nutritional needs of a given strain, all influencing factors should be set and controlled to known conditions, whilst the parameter under study is varied accordingly. For this reason, performing the optimisation process under an optimal laboratory setting is mandatory to guarantee that no external factor is influencing the study at hand.

Some specific examples are:

L69: The study used triplicate examinations of a 1L test system. These numbers are far too low to extrapolate anything on a commercial aquaculture scale (e.g. factors of magnitude higher).

Answer: Given the number of conditions tested, it would not be possible nor feasible to increase the number of biological replicates. As previously mentioned, the bubble columns used in this work have exactly the same diameter and light path as the tubular photobioreactors used at Necton (20 m3 tubular photobioreactors), and the air flow and CO2 injected is adjusted to simulate to the best of our capacity the industrial cultures' growth. Furthermore, 3 replicas per condition tested are sufficient to achieve scientifical relevant results, and it is the standard number of replicates used in most biology and biotechnology works conducted with microalgae. These results correspond to the establishment of the necessary baseline for further scaling up the process and use these diatoms in commercial products.

L81: optics were used for culture growth calculations- no live observations were done? There are numerous issues with this omission. L116-117 says microscopy was done, which showed cell death and elongation- so the density of healthy diatoms could be far lower than presented here.

 Answer: Microscopic observations were conducted daily, and images were added to the manuscript following the reviewers’ suggestions. The results were obtained in the cultures under controlled conditions were silicates proved to be limiting, hence the lower DW’s observed. It is most certain that the presence of healthy diatoms under these conditions was scarce. These results were not observed for the conditions considered optimal, as is stated in the results and discussion. The text was edited to clarify this point for the reviewers and readers of the manuscript.

The conclusions are less than 20 lines, which is far too short. Perhaps this is indicative of the study not being able to conclude anything novel from the simple design?

Answer: Although we do not agree that 20 lines of discussion are short, since most journals require straight to the point conclusions, the conclusions section was rewritten and adjusted to better highlight the findings of the manuscript.

The paper is in perfect British English. The figures are well put together, but the text is confusing jumping between the two species. The choices of what nutrients to include and to what level to vary them are also missing within the text, despite a list of 45 references.

Answer: The authors acknowledge the reference to perfect use of the British English language. For each section of the results, the description and discussion of said results is done separately for each species. Complete separation of the results obtained for each species would lead to a repetitive text as these species had similarities in the results obtained for some of the nutrients optimised. As previously mentioned in the reply for the editor, the authors improved the description of the nutrients that were studied, both in text and with additional figures, to clarify this point for the reviewers and readers of the manuscript. The justification of the choices as referred to by the reviewer can be found in the new manuscript, section 2.1 lines 102-119:

“Through a fed-batch stepwise procedure, several five parameters were tested in triplicate, regarding four concentrations, in the following order: 1) silicates: 0.4, 0.8, 1.2 and 2.4 mM; 2) nitrates: 1, 2, 4 and 8 mM; 3) phosphates: 50, 100, 200 and 400 µM; 4) iron: 10, 20, 40 and 80 µmol; and 5) micronutrients: 0.5, 1-, 2- and 4-mL L-1. Nutribloom plus (NB+) was supplied as control with a final nitrate, phosphate, and iron concentrations of 4 mM, 200 and 40 µM, respectively. Micronutrients and vitamins were provided upon a 1:500 dilution of the respective stock solutions. Silicate was provided at a final concentration of 0.4 mM. The step-wise optimisation approach indicates that each nutrient was optimised one at a time, and the optimised value was used for the following trial. A schematic representation of the trials can be found in Figure 2. The chosen order in nutrient optimisation is based on the existing know-how from previous results obtained by Necton S.A. The experience in growing marine diatoms from lab to industrial scale show a common pattern, where silicate concentration is always the most significant contributor followed by the macronutrients (nitrate and phosphorus) and iron, whereas the micro-nutrient solution commonly has a lower effect (seawater commonly supplies all micronutrients used). Vitamins were not included in the present study, as their use in microalgal commercial scale cultivations is rare due to elevated costs. The composition of the culture medium NB+ is described in Table 1.”

Due to the issues with the methods, rational of the overall study, and lack of novelty (such that most of this work has certainty already been done after a simple google search) I do not feel that this work is suitable for publication in this journal, despite the topic potentially being very helpful to other researchers in the field and overall aquaculture.

Answer: We expect that the answers provided highlighted the novelty and importance of the work performed. In fact, there are no similar works found in the literature for these species, even less in a google search tool. Hence the casual difficulty to provide recent references for most of the results obtained. For this reason, the publication of this work is crucial to assist numerous production units that cannot currently cultivate these species due to high production costs associated with low productivities that do not compensate the original investment.

Reviewer 2 Report

In the reviewed paper interesting and important results on algae biotechnology are presented. The optimization of the culture conditions of high productive diatom algae strains is very important to science and industry. The authors used the proper methodology, the description of all steps of the study is very clear. The figures and the table reflect the results of the investigation.

I recommend the manuscript for publication with minor revision.

Remarks to the authors.

  1. In the introduction you described you give a description of the Skeletonema costatum and Chaetoceros calcitrans. I think it is necessary to add more information about the biology and physiology of these algae
  2. Please, correct the stylistic mistakes:
  • Line 13: products thereof – correct.
  • Line 13: productivities associated with high production – try to find a synonym.
  • Line 20: therefore – try to find a synonym. You use this word very often.
  • Line 52: PUFAs – explain here instead of line 57.

Author Response

Reviewer #2

In the reviewed paper interesting and important results on algae biotechnology are presented. The optimisation of the culture conditions of high productive diatom algae strains is very important to science and industry. The authors used the proper methodology, the description of all steps of the study is very clear. The figures and the table reflect the results of the investigation.

Answer: The authors acknowledge the reviewer for the kind words. 

I recommend the manuscript for publication with minor revision.

Remarks to the authors.

  1. In the introduction you described you give a description of the Skeletonema costatum and Chaetoceros calcitrans. I think it is necessary to add more information about the biology and physiology of these algae

Answer: The introduction section was improved and extended according to the reviewer comment.

  1. Please, correct the stylistic mistakes:
  • Line 13: products thereof – correct.
  • Line 13: productivities associated with high production – try to find a synonym.
  • Line 20: therefore – try to find a synonym. You use this word very often.
  • Line 52: PUFAs – explain here instead of line 57.

Answer: All suggestions were introduced in the manuscript as suggested by the reviewer.

Reviewer 3 Report

The manuscript is an article entitled: ‘Optimisation of Biomass production and Nutritional Value of 2 Two Marine Diatoms (Bacillariophyceae), Skeletonema costatum and Chaetoceros calcitrans'.

The study conducted is very interesting, because the issue of optimising culture media is significant and could actually bring many benefits from an application point of view in the fields of biotechnology, pharmaceuticals, nutraceuticals and so on.
However, before accepting the article, I would recommend that the authors implement the manuscript. Here are some comments and suggestions.

Simple Summary and Abstract - Why was a simple summary also written? I would suggest merging the simple summary and the abstract into one text. For example, in the abstract, I think the content of lines 25-26 is not essential, but I suggest inserting some initial lines to motivate the study, as done at the beginning of the simple summary.

Introduction - I would suggest giving a few more examples from the literature to answer the following questions:
- Are there other examples in the literature of studies to optimise culture media?
- With media having different concentrations and characteristics, what yields have been observed in terms of productivity?
- Why is this study necessary?
- Lines 47-48: "given their outstanding potential for different biotechnological applications"... some examples and references are needed.
- Lines 51-52: "Because of its rich nutritional profile, mainly in lipids, proteins and PUFAs"... same comment as above. Give a few more examples, both qualitative and quantitative, from the literature.

Materials and methods - This section should also be expanded to better explain the various steps and make the protocols reproducible.
- Line 68: there is no reference for Necton's culture collection.
- Line 71-72: why is the initial dry weight of the two cultures different (0.18 vs 0.12)? Reason?
- It is not very clear to me how the different nutrient concentrations were supplied. What is the initial composition of the culture medium? Were the concentrations of the various nutrients varied one by one? How many different combinations of media were obtained? This needs to be better explained and, perhaps, a diagram (e.g. a table) to show the various combinations of nutrients would be useful. To be more precise: when the silicate concentration was 0.4 mM, what was the concentration of the other nutrients? Same logic applies to all combinations.
- Lines 77-78: What are the concentrations of the stock solutions of micronutrients and vitamins? Knowing this is essential, because otherwise you cannot know their final concentration after dilution.
- Lines 84-85: How have the protocols been modified? Can you provide any more information?
- Lines 90-91: How much biomass was collected? How? Is there a specific volume that should be harvested to get enough biomass to run all the following biochemical assays? Otherwise it cannot be properly assessed whether the biomass is sufficient to complete the measurements.
- In section 2.4 I think it would be useful to write down the number of replicates done and also how the results are reported in the Discussion section, i.e. are the values averages ± standard deviations? This is written in the figure captions, but should also be reported here in my opinion.

Results and discussion
- In general, I think it is useful to report some more data from the literature for comparison, otherwise it is not clear whether the growths observed in this study are so much greater than those observed in previous studies. For example: "in work #, at X concentration of nutrient Y a growth of... was observed". This comment applies to the whole discussion section, because I noticed a lack of examples.
- Line 108: insert the reference of the figures where the growths can be observed, because this is the first sentence where they are mentioned.
- Line 116: under what concentration were these observations made? It is not clear.
- Line 155: What specifically was seen with the microscope observations? Explain a little further.
- Lines 185-186-187: develop this part further by giving examples.

Conclusions
I have a question for the authors. After all the conditions tested, have they identified the optimal composition that the medium should have - in terms of all the nutrients - to achieve the best results? If yes, I would write this in the discussion or in the conclusions.

After these modifications, which I think are useful to improve the quality of the work done and make it more reproducible, I would certainly suggest accepting the work, which would be extremely useful.

Author Response

Reviewer #3

The manuscript is an article entitled: ‘Optimisation of Biomass production and Nutritional Value of 2 Two Marine Diatoms (Bacillariophyceae), Skeletonema costatum and Chaetoceros calcitrans’.

The study conducted is very interesting, because the issue of optimising culture media is significant and could actually bring many benefits from an application point of view in the fields of biotechnology, pharmaceuticals, nutraceuticals and so on.
However, before accepting the article, I would recommend that the authors implement the manuscript. Here are some comments and suggestions.

Answer: The authors acknowledge the reviewer for the kind words.

Simple Summary and Abstract - Why was a simple summary also written? I would suggest merging the simple summary and the abstract into one text. For example, in the abstract, I think the content of lines 25-26 is not essential, but I suggest inserting some initial lines to motivate the study, as done at the beginning of the simple summary.

Answer: The inclusion of a simple summary is requested by the Biology journal, as can be seen in the manuscript template for the biology journal. The purpose of the simple summary is to explain the work in a simple and concise manner to the readers.

Introduction - I would suggest giving a few more examples from the literature to answer the following questions:
- Are there other examples in the literature of studies to optimise culture media?

Answer: The introduction section was improved following the suggestions of the different reviewers and editor. To the best of our knowledge, all studies (4 references) were cited in the present manuscript.

- With media having different concentrations and characteristics, what yields have been observed in terms of productivity?

Answer: Unfortunately, the authors could not find comparable information in the literature, since the system conditions always differed or were not specified by the authors.

- Why is this study necessary?

Answer: As previously mentioned in the reply for reviewer #1, the importance and rationale of the study was better defined in the manuscript to clarify the doubts raised by the reviewers and future readers. For example, a sentence was added at the end of the introduction to highlight the relevance of the work conducted: “ … which is crucial for reduced production costs associated with higher productivities.”

- Lines 47-48: “given their outstanding potential for different biotechnological applications”... some examples and references are needed.

Answer: Following the reviewer suggestion, additional references were added.

- Lines 51-52: “Because of its rich nutritional profile, mainly in lipids, proteins and PUFAs”... same comment as above. Give a few more examples, both qualitative and quantitative, from the literature.

Answer: Further examples from the literature were given according to the reviewer's comment.

Materials and methods - This section should also be expanded to better explain the various steps and make the protocols reproducible.
- Line 68: there is no reference for Necton’s culture collection.

Answer: Information was added accordingly.

- Line 71-72: why is the initial dry weight of the two cultures different (0.18 vs 0.12)? Reason?

Answer: This work was implemented alongside the production unit of the company Necton S.A. For matters of comparison, the initial optical density implemented was the same used by the company, which for both species corresponded to 0.3. However, since these species are morphologically different, distinct DWs were obtained, with Skeletonema costatum having a higher value given its greater size. Information was added to clarify this matter.

- It is not very clear to me how the different nutrient concentrations were supplied. What is the initial composition of the culture medium?

Answer: The information regarding the different nutrient concentrations was added as well as the initial composition of the culture medium.

Were the concentrations of the various nutrients varied one by one?

Answer: The optimisation was performed as a step-wise process, where each nutrient was optimised separately, with 4 different concentrations being tested simultaneously. We altered this section to clarify this matter.

How many different combinations of media were obtained? This needs to be better explained and, perhaps, a diagram (e.g. a table) to show the various combinations of nutrients would be useful. To be more precise: when the silicate concentration was 0.4 mM, what was the concentration of the other nutrients? Same logic applies to all combinations.

Answer: We agree with reviewer and a table was added to better clarify the combinations and concentrations of the nutrients used in the different trials and treatments.

- Lines 77-78: What are the concentrations of the stock solutions of micronutrients and vitamins? Knowing this is essential, because otherwise you cannot know their final concentration after dilution.

Answer: The information suggested was added to the main manuscript.

- Lines 84-85: How have the protocols been modified? Can you provide any more information?

Answer: The reference with the modifications used was added to the manuscript.

- Lines 90-91: How much biomass was collected? How? Is there a specific volume that should be harvested to get enough biomass to run all the following biochemical assays? Otherwise it cannot be properly assessed whether the biomass is sufficient to complete the measurements.

Answer: The biochemical analysis was conducted in the final trial, and whole culture was harvested, freeze dried, and stored for biochemical analysis. The analyses were conducted using methods optimised during the last 12 years, and require very low quantities of biomass. The manuscript text was adjusted as suggested.

- In section 2.4 I think it would be useful to write down the number of replicates done and also how the results are reported in the Discussion section, i.e. are the values averages ± standard deviations? This is written in the figure captions, but should also be reported here in my opinion.

Answer: The authors agree with the reviewer and the information was altered as suggested.

Results and discussion
- In general, I think it is useful to report some more data from the literature for comparison, otherwise it is not clear whether the growths observed in this study are so much greater than those observed in previous studies. For example: “in work #, at X concentration of nutrient Y a growth of... was observed”. This comment applies to the whole discussion section, because I noticed a lack of examples.

Answer: This information was altered to better explain this. However, this was not always possible to achieve due to lack of information available in the literature. This lack of information translates into incomplete methodological descriptions, or even different growth measurements methods that were applied, or incomparable works due to the different growth systems and testing conditions used. For these reasons, we focused on performing a literature review to describe the highest growths (depicted as dry weights, for both diatom species, as you can find in section 3.1.1, where we demonstrate that we achieved significantly higher growth parameters with our system and optimisation of culture medium. For these reasons, we also chose to focus on the nutrient concentrations reported in the literature, for these two species, since this was the focus of our work

- Line 108: insert the reference of the figures where the growths can be observed, because this is the first sentence where they are mentioned.

Answer: The information was added as suggested.

- Line 116: under what concentration were these observations made? It is not clear.

Answer: The information was added as suggested.

- Line 155: What specifically was seen with the microscope observations? Explain a little further.

Answer: This information was added as suggested.

- Lines 185-186-187: develop this part further by giving examples.

Answer: Unfortunately, the authors could not find any studies regarding the effects of micronutrients on these two diatom species, in terms of optimal concentrations. As the authors mention in lines 187-188, the studies that do exist are related to the toxic effects that micronutrients such as zinc and cobalt can have.

Conclusions
I have a question for the authors. After all the conditions tested, have they identified the optimal composition that the medium should have - in terms of all the nutrients - to achieve the best results? If yes, I would write this in the discussion or in the conclusions.

Answer: The work significantly improved the basal medium used by Necton, and key nutrients were almost fully optimised. This information was added to the Results & Discussion section.

After these modifications, which I think are useful to improve the quality of the work done and make it more reproducible, I would certainly suggest accepting the work, which would be extremely useful.

-----

Reviewer 4 Report

This paper is a good example of transferring research to the industry. The main contribution of the paper is that certain culture parameters can be the same for both species of diatoms, which simplify the culturing techniques, at the same time that high growth rates are obtained. I wish to comment that the authors have been worried about using recent studies as references, which is always the best practice and the clear and well-structured paper that the authors have submitted.

In the paper, some aspects remain unclear. The authors have selected two diatoms species from a culture collection, but there is no ID code or reference about the species used. It is usual that in a culture collection species are replicated but they don’t share the same code, so it will be desirable that the authors reflect this fact briefly on the paper. Also, more information about micronutrients and vitamins stocks should be provided, at least as supplementary material to replicate the methodology applied with other species. 

Despite the use of these two species for the obtention of high-value marine bioproducts is not new, the authors go deeper into the optimization of culturing media composition regarding the protein, carbohydrates and fatty acids production rates. The rates obtained are above other studies previously reported, mainly focused on fatty acid production, which makes the paper relevant in this field. References used are appropriate and current and an abnormal number of self-citation haven’t been observed. In addition, the conclusions are well supported by the references and by the figures and tables shown in the paper.

On the other hand, reviewing the paper, some specific aspects that should be clarified by the authors. The suggestions are:

    • Add the ID codes of the species used in the study, according to the collection database (line 68).
    • If a commercial stock of micronutrients and vitamins was used, please mention it. If not, it will be desirable that the authors detailed the composition and the concentrations as supplementary material (line 77).
    • In the results and discussions part there is no mention of vitamins, are they relevant or not? Please, add a comment.
    • In the figures included by the authors, the error bars are not always easily seen. Could the authors include in a table all the data obtained? Maybe as supplementary material. They have included a similar table with the FA concentrations. 
    • In 3.1.2. section, the authors suggest that 4 mM is the ideal concentration of nitrates that should be added for culturing both species but, if Figure 1 is observed, 2 mM seems to be more effective after 4 days of culture. Can the authors explain why they decided to address 4 mM as the ideal concentration for S. costatum?

Author Response

Reviewer #4

This paper is a good example of transferring research to the industry. The main contribution of the paper is that certain culture parameters can be the same for both species of diatoms, which simplify the culturing techniques, at the same time that high growth rates are obtained. I wish to comment that the authors have been worried about using recent studies as references, which is always the best practice and the clear and well-structured paper that the authors have submitted.

Answer: The authors acknowledge the reviewer for the kind words.

In the paper, some aspects remain unclear. The authors have selected two diatoms species from a culture collection, but there is no ID code or reference about the species used. It is usual that in a culture collection species are replicated but they don’t share the same code, so it will be desirable that the authors reflect this fact briefly on the paper. Also, more information about micronutrients and vitamins stocks should be provided, at least as supplementary material to replicate the methodology applied with other species. 

Answer: The ID code of both strains was added to the manuscript. Also, information regarding the micronutrients was added.

Despite the use of these two species for the obtention of high-value marine bioproducts is not new, the authors go deeper into the optimisation of culturing media composition regarding the protein, carbohydrates and fatty acids production rates. The rates obtained are above other studies previously reported, mainly focused on fatty acid production, which makes the paper relevant in this field. References used are appropriate and current and an abnormal number of self-citation haven’t been observed. In addition, the conclusions are well supported by the references and by the figures and tables shown in the paper.

On the other hand, reviewing the paper, some specific aspects that should be clarified by the authors. The suggestions are:

    • Add the ID codes of the species used in the study, according to the collection database (line 68).

Answer: This information was added to manuscript as indicated previously.

    • If a commercial stock of micronutrients and vitamins was used, please mention it. If not, it will be desirable that the authors detailed the composition and the concentrations as supplementary material (line 77).

Answer: This information was added to manuscript as indicated previously.

    • In the results and discussions part there is no mention of vitamins, are they relevant or not? Please, add a comment.

Answer: The absence of vitamins in our work was a conscious decision, explained by the absence of vitamins in commercial culture media due to high costs that would be prohibitive for large volumes. Vitamins are only used in commercial scale ventures for microalgal strains whose biomass has a very high commercial value.

    • In the figures included by the authors, the error bars are not always easily seen. Could the authors include in a table all the data obtained? Maybe as supplementary material. They have included a similar table with the FA concentrations.

Answer: The error bars from the growth graphs are not easily seen due to very low standard deviations between replicates. A table was added as supplementary material as suggested by the reviewer.

    • In 3.1.2. section, the authors suggest that 4 mM is the ideal concentration of nitrates that should be added for culturing both species but, if Figure 1 is observed, 2 mM seems to be more effective after 4 days of culture. Can the authors explain why they decided to address 4 mM as the ideal concentration for S. costatum?

Answer: The reviewer is correct. Unfortunately, the image had an error where the graphs a and b were swapped. The plots were corrected accordingly.

Round 2

Reviewer 1 Report

The new images are excellent, but of course I do not agree with your replies to my suggestions. I can see however that the new draft is a rewrite. Since my first comments largely were not addressed, I have nothing further to add.